# Stress-shape misalignment in confluent cell layers

**Mehrana R. Nejad** [1] ✉, **Liam J. Ruske**[1], **Molly McCord** [2,3], **Jun Zhang**[2,4], **Guanming Zhang** [5,6], **Jacob Notbohm** [2,3,4] ✉ **& Julia M. Yeomans** [1] ✉

In tissue formation and repair, the epithelium undergoes complex patterns of motion driven by the active forces produced by each cell. Although the principles governing how the forces evolve in time are not yet clear, it is often assumed that the contractile stresses within the cell layer align with the axis defined by the body of each cell. Here, we simultaneously measured the orientations of the cell shape and the cell-generated contractile stresses, observing correlated, dynamic domains in which the stresses were systematically misaligned with the cell body. We developed a continuum model that decouples the orientations of contractile stress and cell body. The model recovered the spatial and temporal dynamics of the regions of misalignment in the experiments. These findings reveal that the cell controls its contractile forces independently from its shape, suggesting that the physical rules relating cell forces and cell shape are more flexible than previously thought.

Cells are the fundamental building blocks of life, and their ability to collectively generate active forces plays a crucial role in physiological processes from morphogenesis[1], tissue growth and repair[2] to apoptosis[3], tumour development[4] and metastasis[5]. Confluent cell monolayers plated on adhesive substrates are widely used as model systems in investigations aimed at understanding collective cell motility[6,7]. There is growing evidence that the dynamics of such confluent cell layers can often be well described by the theories of active nematics[8–12].

Describing cells as active emphasises that they continuously take energy from their surroundings and use it to initiate life processes[13,14]. Nematic particles are elongated in shape, and nematic ordering occurs when their long axes tend to align parallel (Fig. 1a). Such ordering has frequently been observed in long, thin cells such as fibroblasts[8,15] or LP-9[16] but is more surprising in cell types that are on average isotropic, such as the Madin-Darby Canine Kidney (MDCK) cell line. Here extensions in cell shape, driven by active forces, are locally correlated to give nematic order[17–19].

The combination of activity and nematic ordering leads to striking collective behaviours, which are mirrored between active nematic models and cell monolayers. These include active turbulence, characterised by cell velocities that are chaotic with regions of high vorticity[20], spontaneous directed flow in confinement[21], and the identification of motile topological defects[18], long-lived cell configurations at which domains of different cell orientations meet and the nematic order is frustrated. Figure 1b, c show the cell orientations around the +1/2 and −1/2 defects which predominate in 2D cell layers.

The direction of the long axis of a cell is an obvious way to define a preferred nematic shape axis. The local axis of principal stress gives an alternative way to choose the local axis of any nematic ordering. In almost all work to date, it has been assumed that the two definitions are equivalent, meaning that the stress and shape axes are tightly coupled[17,22–24]. Under this assumption, differences between the axes of stress and shape would be modest and attributable to biological noise. Here we challenge this assumption and show that there are dynamic, correlated regions in cell layers, where the stress and shape axes are

[1]The Rudolf Peierls Centre for Theoretical Physics, Department of Physics, University of Oxford, Parks Road, Oxford OX1 3PU, United Kingdom. [2]Biophysics Program, University of Wisconsin–Madison, Madison, WI, USA. [3]Department of Mechanical Engineering, University of Wisconsin–Madison, Madison, WI, USA. [4]Department of Engineering Physics, University of Wisconsin–Madison, Madison, WI, USA. [5]Center for Soft Matter Research, Department of Physics, New York University, New York, NY 10003, USA. [6]Simons Center for Computational Physical Chemistry, Department of Chemistry, New York University, New York, NY 10003, USA. ✉e-mail: mehrana@g.harvard.edu; jacob.notbohm@wisc.edu; julia.yeomans@physics.ox.ac.uk

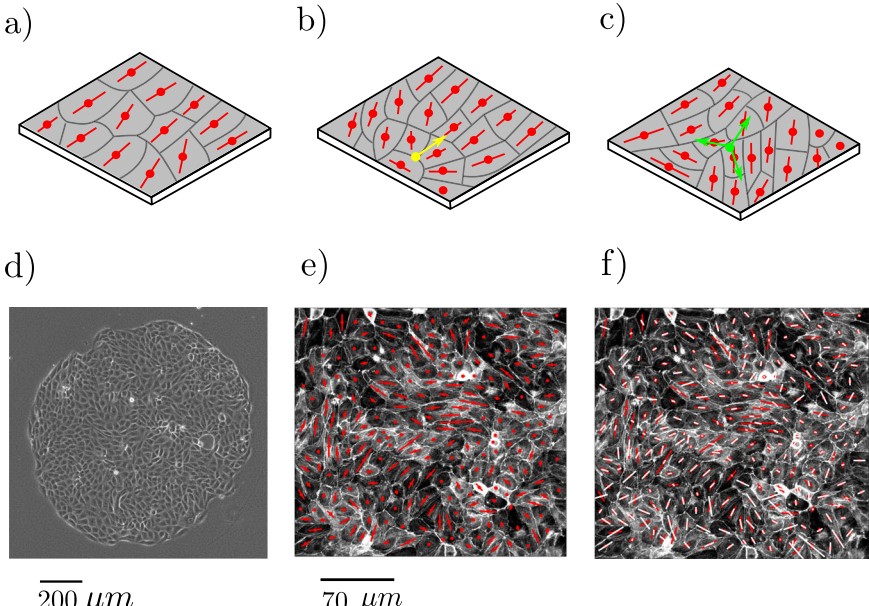

**Fig. 1 | Snapshots of cells in the layer. a** Nematic ordering of cell shape in a confluent cell monolayer. Red lines indicate the nematic directors **n**, which lie along the long axis of a cell, and which tend to align parallel. **b** Cell shape orientations around a +1/2 topological defect are shown in yellow. **c** Cell shape orientations around a −1/2 topological defect are shown in green. **d** Geometry of the MDCK monolayers used in the experiments. **e** Example of a typical experiment measuring the long axis of cell shape orientation, **n**, (red lines) in an MDCK monolayer. **f** A typical set of results from experiments measuring the orientation of maximum contractile stress **m** (white lines) overlaid on the orientations of cell shape, **n** (red lines). In (**e**–**f**), the lengths of the red (white) lines are proportional to the deviation of the cell aspect ratio from unity (magnitude of the contractile stress). Experiments are repeated independently 11 times with similar results.

systematically misaligned. Introducing the possibility of such misalignment in a continuum model of active nematics reproduces the temporal and spatial correlations and misalignment angle observed in our experiments and emphasises the key role of activity in driving the misalignment.

## Results

Our experiments were performed on confluent MDCK layers of diameter 1 mm plated on polyacrylamide substrates with Young's modulus of 6 kPa (Fig. 1d). We define the cell shape orientation in the tissue by assigning each cell a director (headless vector) **n** which lies along the long axis of the anisotropic cell shape as shown in Fig. 1e[25]. Monolayer Stress Microscopy was used to measure the stress tensor **σ**, from which we computed the orientation of the first principal stress, which defines the local orientation **m** along which contractile forces are generated (Fig. 1f). By interpolating between individual cells we obtain continuous director fields **n** and **m** which respectively describe cell shape orientation and the principal axis of contractile stress throughout the tissue (see Methods).

We analysed the local cell shape and stress orientations over the course of 12 h in 11 MDCK tissue samples, in time-lapse experiments taking data every 15 min. We define $\theta$ as the misalignment angle between the local cell shape orientation **n** and the principal axis of contractile stress **m** in the tissue (Fig. 2a). The distribution of $\theta$ is shown in Fig. 2b. While most cells create contractile stresses along their cell shape axis ($\theta \approx 0$), there is a large number of cells in which the axis of contractile stress is significantly misaligned with respect to shape orientation. If the misalignment angle reaches $\theta \approx 90°$, cells create contractile stresses perpendicular to the cell shape orientation, thereby pulling inward not along their long shape axis but rather along their short shape axis. In the following, we will refer to cells with large misalignment ($\theta > 45°$) as extensile and cells with small misalignment ($\theta < 45°$) as contractile following the usual terminology in the mathematics and active matter literature, and we distinguish these ranges of $\theta$ as blue and red in Fig. 2b.

We now investigate the spatial and temporal correlation of the shape and the stress orientations in the MDCK monolayers. Figure 2c (left) shows a tissue snapshot where the cell shape orientation field **n** is shown as black lines, and the colour map again indicates whether $\theta$ is greater (extensile, blue) or less (contractile, red) than 45°. Figure 2c (right) shows the time evolution of a region of the tissue with snapshots taken at 15-min intervals. Similar data is presented dynamically in Supplementary Movie 1. It is evident that the misalignment angle forms evolving spatiotemporal patterns where extensile cells form small, dynamic clusters in a mostly contractile background. The extensile clusters grow, shrink and coalesce over time. The time-averaged area fraction of extensile cells is 27 ± 4% (Fig. 2d).

To further quantify the spatial patterns, we calculated the spatial and time correlation functions of the cell shape orientation, the cell stress orientation, and the mismatch angle $\theta$. These are defined as

$$C^x(r) = \langle \cos 2(\psi_x(r+r_0,t_0) - \psi_x(r_0,t_0)) \rangle_{t_0,r_0}, \quad C^x(t) = \langle \cos 2(\psi_x(r_0,t+t_0) - \psi_x(r_0,t_0)) \rangle_{t_0,r_0},$$
(1)

where $\psi_x$ represents the shape director angle, stress angle, or the mismatch angle $\theta$, and $\langle \ldots \rangle_{r_0,t_0}$ denotes an average over space (a circle of diameter 312 μm in the centre of the island to avoid edge effects) and time. The spatial correlations, shown in Fig. 3a, indicate a length-scale ~50 μm for the cell stress orientation, and a longer length-scale ~100 μm for the cell shape orientation. From the time correlation function in experiments, in Fig. 3b, we identify a time scale for the decay of the extensile patches ~300 min.

Many cells contain bundles of actomyosin, termed stress fibres, that tend to form along the long axes of cells and are the primary source of contractile stresses[26]. We hypothesise that the regions of large misalignment angle are due to active flows disturbing the natural alignment of the stress fibres with the long axis of the cell due to different responses of the shape axis **n** and the stress axes **m** to flows. This leads to the formation of extensile regions which have a large

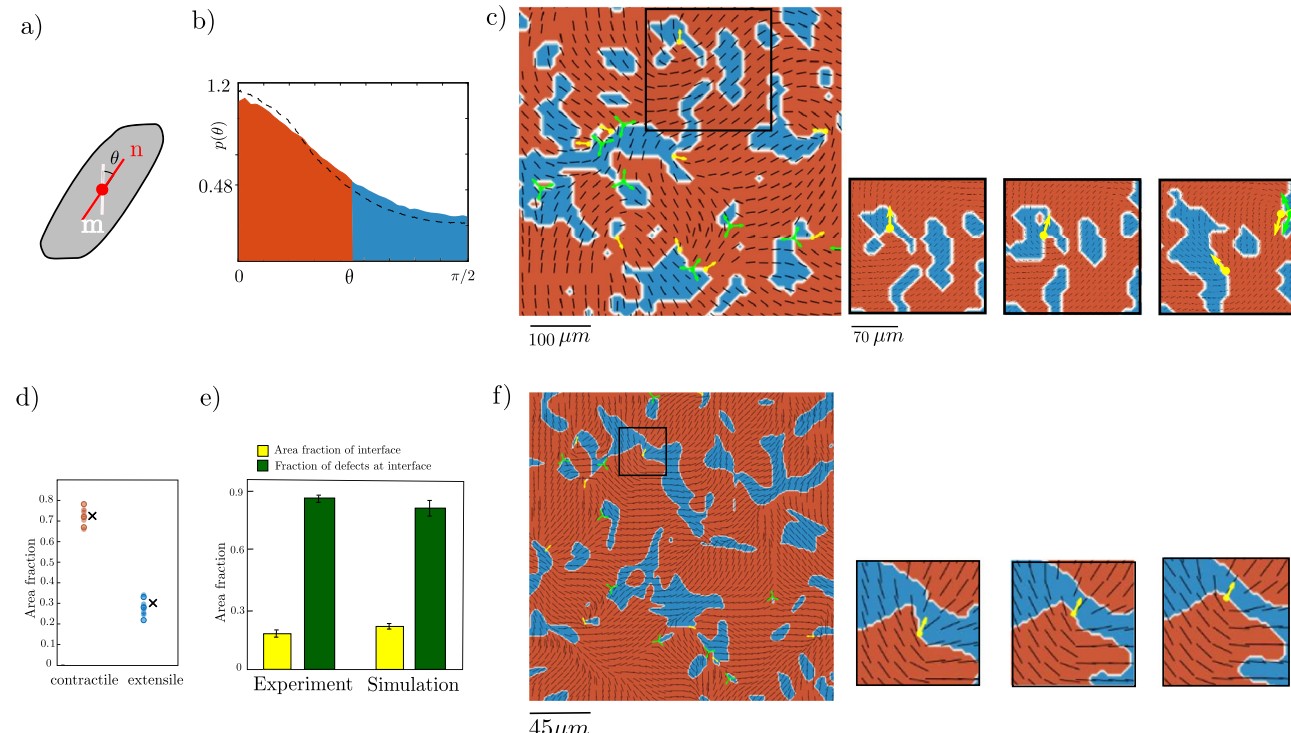

**Fig. 2 | Misalignment between cell orientation and stress in experiments and simulations. a** Definition of the misalignment angle $\theta$ between the shape orientation axis **n**, and the principal axis of contractile stress **m**. **b** Distribution of the misalignment angle $\theta$. Red/blue colouring denotes contractile ($\theta < 45^o$)/extensile ($\theta > 45^o$) values. The distribution contains data from 11 independent MDCK islands and all time points. The black dotted line shows the result from the simulation. **c** Left: Tissue snapshot with the cell orientation field **n** shown as black lines on top of a colour map distinguishing contractile (red) and extensile (blue) regions. Topological defects in the cell orientation are indicated by yellow (+1/2) and green (−1/2) symbols. Right: Snapshots of a region of the same tissue taken 15 min apart showing the evolution of extensile clusters. The time axis is from left to right. **d** Experimental time average of the area fraction of contractile (red) and extensile (blue) cells. The crosses show the results from simulations. The time-averaged area fraction of extensile cells is 27 ± 4% (28 ± 2%) in experiments (simulations).

**e** Defects are preferentially found in the vicinity of boundaries between extensile and contractile regions. A spatial and temporal average on all frames of the experiments and simulations show that interface domains make up about 18 ± 2% of the tissue area fraction in the experiments and about 21 ± 3% of the area fraction in simulations. However, 86 ± 2% (82 ± 7.9%) of defects lie within interface domains in experiments (simulations). In (**d**, **e**) the experimental data shows the behaviour over 11 different experiments over 12 h (48 data points in each experiment). The simulation data is calculated over all frames of the simulations after the steady state is reached (150 data points). The error bars represent the standard deviation. **f** Snapshot from simulations with the cell orientation field **n** shown as black lines on top of a colour map distinguishing contractile (red) and extensile (blue) regions. Topological defects in the cell orientation are indicated by yellow arrow (+1/2) and green trefoil (−1/2) symbols. The time axis is from left to right. Source data are provided as a Source Data file in ref. 45.

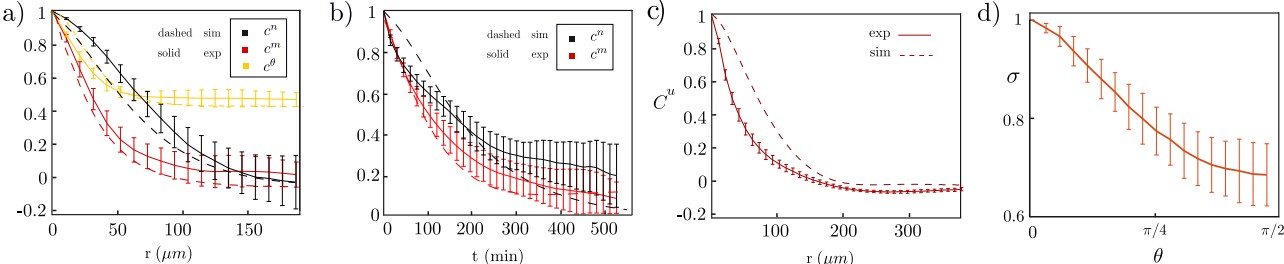

**Fig. 3 | Spatial and temporal correlations in experiment and simulations. a** Decay of the spatial correlation functions, $C^x(r) = \langle \cos 2[\psi_x(r+r_0,t_0) - \psi_x(r_0,t_0)]\rangle_{t_0,r_0}$, where $\psi_x$ represents shape director angle $C^\mathbf{n}$, stress angle $C^\mathbf{m}$, or the mismatch angle $C^\theta$. Comparison of the simulation and the experiment assumes $10 LB$ spatial units ~30 μm. **b** Decay of the time correlation functions, $C^x(t) = \langle\cos 2[\psi_x(r_0,t+t_0) - \psi_x(r_0,t_0)]\rangle_{t_0,r_0}$, where $\psi$ represents shape director angle $C^\mathbf{n}$ or the stress angle $C^\mathbf{m}$. Comparison of the simulation and the experiment assumes $100 LB$ time units ~10 min. **c** Using the length-scale that relates experiments and simulations (in part (**a**)), we can compare the velocity correlation in

experiments (solid line) and simulations (dashed line). The velocity correlation in simulations decays on a similar length-scale as in the experiments. The velocity correlation and the error bars are measured in eight experiments. **d** Magnitude of the stress $\sigma$ as a function of the misalignment angle $\theta$. The magnitude of the stress is re-scaled with its maximum value in each experiment before averaging. In (**a**–**d**) the centre of the error bars show the average data and the error bars show the standard deviation over 11 experiments. The data in each experiment is averaged over 48 frames. Source data are provided as a Source Data file in ref. 45.

mismatch between the shape and stress axes. A cell's stress axis and shape axis then gradually relax towards each other.

Continuum tissue models, based on the equation of motion of active nematics, have been very successful in explaining cell motility on a coarse-grained level[8–11,18]. However, the assumption has always been that the principal axis of contractile stress **m** and the shape axis **n** are indistinguishable. Therefore, to investigate the consequences of our hypothesis, we extend the continuum modelling to decouple **m** and **n**.

We describe the shape of the cells and the stress using nematic order parameters $\mathbf{Q}^n = S^n(\mathbf{n} \otimes \mathbf{n} - \mathbf{I}/2)$ and $\mathbf{Q}^m = S^m(\mathbf{m} \otimes \mathbf{m} - \mathbf{I}/2)$, respectively[25]. The nematic order parameters encode the magnitude of nematic order in cell shape $S^n$ or in the stress $S^m$, and the director field associated with cell shape, **n** or the stress **m**. We assume that the flows are created by contractile active forces that act along the direction of stress fibres **m**. The active flows advect the cells, and the vorticity of the flows rotates both the shape and the stress director fields. The experimental spatial correlation functions show that the nematic order of the shape director **n** has a longer length-scale than that of the stress director **m**, which we model by choosing different elastic constants in the free energy. This also means that the shape and the stress directors respond in different ways to the vortical flows leading to misalignment between **m** and **n**. We include a term in the free energy which acts to slowly relax the misalignment. In the continuum simulations, we have the freedom to choose a length and time scale, and we do this by matching the length and time scales of the correlation functions (Eq. (1)) between simulation and experiment, as shown in Fig. 3a, b. See Methods for further details of the modelling.

We compare the simulation results to the experiments in Fig. 2. In agreement with the experiments, spatially correlated domains of extensile cells in a contractile background emerge in the simulations (Fig. 2f and Supplementary Movie 2). We also obtain a quantitative match to the probability distribution for the misalignment angle $\theta$ if we use a realignment time scale of 25 min (Fig. 2b). It is also interesting to compare the velocity-velocity correlation function

$$C^{\mathbf{u}}(r) = \langle \mathbf{u}(r + r_0, t_0) \cdot \mathbf{u}(r_0, t_0) \rangle_{r_0, t_0} / \langle \mathbf{u}(r_0, t_0) \cdot \mathbf{u}(r_0, t_0) \rangle_{r_0, t_0} \qquad (2)$$

measured in experiments and calculated in simulations while maintaining the same mapping between experimental and simulation parameters. Figure 3c shows a very reasonable agreement of the decay length-scale.

Active turbulence is also characterised by topological defects, and we noticed that topological defects in the cell orientation field tend to sit at the interfaces between extensile and contractile domains. This is illustrated in Fig. 2c (experiment) and f (simulations), where +1/2 defects are indicated by a yellow arrow and −1/2 defects by a green trefoil. To quantify the results, we consider the points that are at a distance smaller than $r < 5.2$ μm (equal to the spatial resolution of the stress measurement) from the interface as the interface domain, while the rest of the tissue is defined as the bulk domain. A spatial and temporal average shows that interface domains make up about $28 \pm 2\%$ of the tissue area fraction in the experiments, and about $21 \pm 3\%$ of the area fraction in simulations. However, about $86 \pm 2\%$ ($82 \pm 7.9\%$) of defects lie within interface domains in experiments (simulations). These results are shown in Fig. 2e.

We emphasise that the exact mechanism for the director dynamics depends on biological detail and is complex. Therefore any quantitative match to our minimal model is due to a somewhat arbitrary tuning of parameters and should only be used to indicate that the mechanism is feasible. We obtain the extensile patches seen in the experiment for a wide range of model parameters and assumptions, see for example Fig. S1 in the SI.

To create extensile islands, the shape and stress dynamics must respond differently to the active stress. In the model this is provided by the different elastic constants which cause a different response to the activity. The experimental evidence behind this assumption is the different length-scales for the decays of the spatial correlation functions of the stress and the shape directors (Fig. 3a). The model assumes that the flows rotate the directors to change the cells from contractile to extensile. This could also happen in the experiments, but another possibility is that the cells shrink to circular and then re-extend in a perpendicular direction. A visual inspection of the data suggests that the transition is mostly happening by rotation rather than deformation. To check this more quantitatively, we measured the cell aspect ratio as regions changed from contractile to extensile (see Fig. S2 in the SI). The results show some change in cell shape during the transition, but it is not large, indicating cell rotation is more likely, which is consistent with the assumptions of the model.

Our interpretation leads to the prediction that in the extensile regions, the stress fibres will be in the process of re-forming to realign with the new direction of cell elongation and are, therefore, less efficient in producing contraction. In Fig. 3d, we plot the magnitude of the contractile stress as a function of $\theta$, showing a clear decrease. Moreover, to check our interpretation visually, we fixed cells for fluorescent imaging of actin fibres in a selection of samples. We found that it was rarely possible to visually ascertain an unambiguous direction of the stress fibres in cells with large $\theta$, whereas stress fibres were much clearer in cases when they were aligned along the long axis of the cell (see SI, Fig. S3). We have also measured the cell aspect ratio in the contractile and extensile regions (SI, Fig. S2c). There is a small reduction in the extensile regions which is consistent with these arguments.

We next, as a comparison, performed similar experiments on the human mesothelial cell line LP-9 (Fig. 4a). These cells, which have an elongated morphology with a high aspect ratio, showed a behaviour that contrasted with the MDCK islands. A very small number of topological defects were present at the beginning of the experiments. These persisted and remained in approximately the same position throughout the experiments (40 h), and no new defects were created, indicating that the cell layer was behaving primarily as a passive nematic, with any active flows not sufficiently strong to create defects or substantially change the cell orientation[8].

There was, however, still a population of extensile cells with $\theta > 45°$, but it was far smaller than in the MDCK monolayers. The extensile cells formed small (3.4% of the area of the tissue) clusters (Fig. 4b) adjacent to the few remaining defects. The time correlations of cell shape and stress were virtually constant (SI, Fig. S4) and there was no evidence that the remaining defects or associated extensile regions disappear within the time scale of the experiment.

As the shape director **n** does not change we modelled the LP-9 cells by fixing a defect in cell shape **n** at the centre of a circular cellular island and allowed the stress field **m** to relax. An extensile region was indeed formed next to the defect in cell shape **n**, as shown in Fig. 4c. This dynamical steady state is a result of the balance between the elastic energy, which favours nematic alignment of the stress directors **m**, and the term which encourages **m** to align with **n**. The exact position and size of the extensile region relative to the defect varies, depending on the initial condition for **m**, indicating the existence of metastable solutions.

## Discussion

The results show that cells in a monolayer do not have the fixed property of exerting force along the cell elongation axis, but change the orientation of their active force to accommodate cell flows and stresses applied by their neighbours. The picture is thus notably more complex than previously thought–our study has revealed an additional internal degree of freedom, the orientation of cell force production. Our work brings a new perspective to the important question of how stress fibres in neighbouring cells align. Moreover, stress-shape misalignment could be relevant for key processes in tissue development

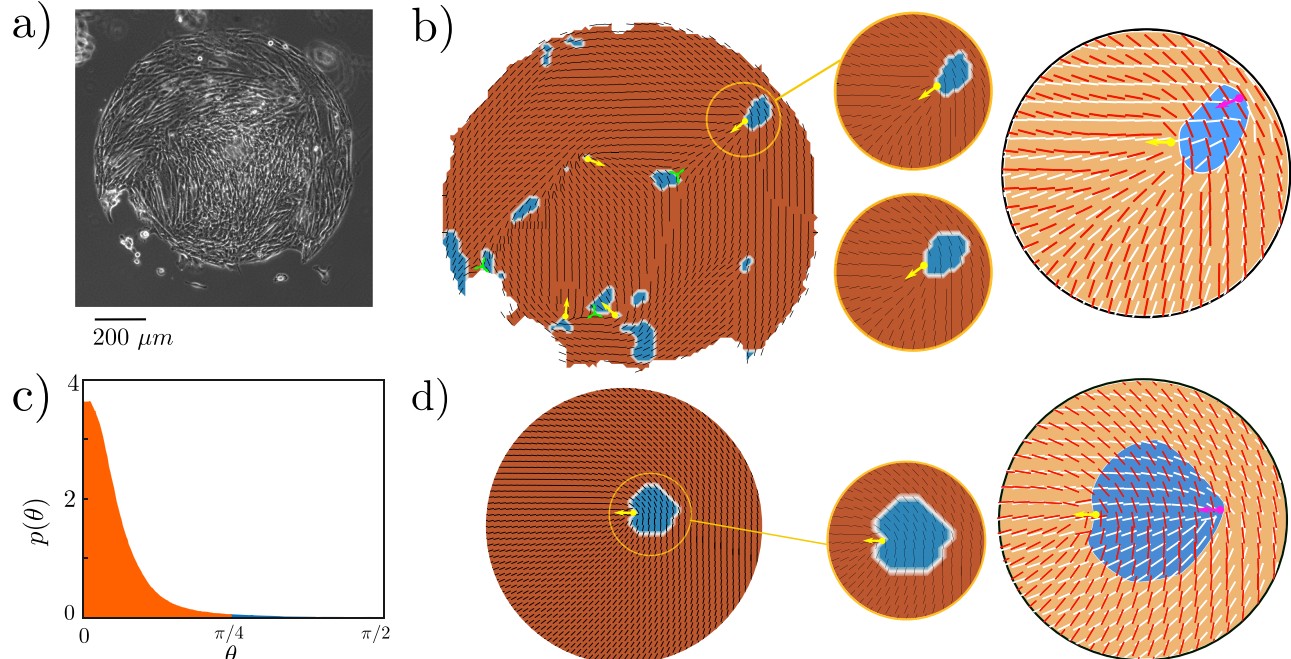

**Fig. 4 | Stress-cell misalignment in LP-9 monolayers. a** Snapshot of an LP-9 island. The experiments on LP-9 cells are repeated 4 times with similar results. **b** left: Tissue snapshot with the cell shape orientation field **n** shown as black lines on top of a colour map distinguishing contractile (red) and extensile (blue) regions. Topological defects in the cell shape orientation are indicated by yellow (+1/2) and green (−1/2) symbols. Middle: Snapshots of a region of the same tissue taken 40 min apart showing that the extensile cluster remains fixed. The time axis is from top to bottom. Right: Close-up view of the same tissue. Cell shape orientation is shown in red and stress orientation in white. The color map shows the extensile (blue) and contractile (orange) regions (smoothed). The defect in the cell shape (stress) orientation is shown in yellow (magenta). **c** Distribution of the misalignment angle θ. Red/blue colouring denotes contractile (θ < 45º)/extensile (θ > 45º) values. The LP-9 cells predominantly form contractile regions (compared with Fig. 2b for MDCK cells). **d** Left: Snapshot from simulations with the cell shape orientation **n** shown as black lines on top of a colour map distinguishing contractile (red) and extensile (blue) regions. The +1/2 defect in the cell shape is indicated by a yellow arrow. Right: Close-up view of the same tissue. Cell shape orientation is shown in red and stress orientation is white. The color map shows the extensile (blue) and contractile (orange) regions (smoothed). The defect in the cell shape (stress) orientation is shown in yellow (magenta). Source data are provided as a Source Data file[45].

and repair and could spur the design of novel active materials having tunable morphology.

## Methods

### Cell culture

Madin-Darby canine kidney (MDCK) type II cells transfected with green fluorescent protein (GFP) attached to a nuclear localisation signal (a gift from Professor David Weitz, Harvard University) were maintained in low-glucose Dulbecco's modified Eagle's medium (10-014-CV, Corning Inc., Corning, NY) with 10% foetal bovine serum (FBS, Corning) and 1% G418 (Corning). LP-9 cells (Coriell, AG07086) were maintained in a 1:1 ratio of Medium 199 and Ham F-12 (Corning) with 10 ng/ml epithelial growth factor (MilliporeSigma) and 0.4 $\mu$g/ml hydrocortisone (MilliporeSigma) that was supplemented with 15% FBS. All cells were maintained in an incubator at 37 °C and 5% $CO_2$.

### Time-lapse imaging and analysis

Polyacrylamide gels embedded with fluorescent particles (0.036% weight/volume 580/605, diameter 0.5 $\mu$m, carboxylate modified; Life Technologies) were fabricated with Young's moduli of 6 kPa and thickness of 150 µm using methods described previously[27,28]. Polydimethylsiloxane (PDMS, Sylgard 184) was cured in 200-µm thick sheets. The sheets were cut into 20 × 10 mm squares and then 1 mm holes were cut using a 1 mm biopsy punch. The PDMS masks were adhered to the gels using previous methods[27], and the 1 mm circular hole was coated with 0.01 mg/ml type I rat tail collagen I (BD Biosciences) with the covalent crosslinker sulfo-SANPAH (Pierce Biotechnology). MDCK and LP-9 cells were seeded onto 1 mm islands 24 h before imaging. The cells and particles were imaged using an Eclipse

Ti-E microscope (Nikon, Melville, NY) with a 10 × numerical aperture 0.5 objective (Nikon) and an Orca Flash 4.0 digital camera (Hamamatsu, Bridgewater, NJ) running on Elements Ar software (Nikon). All imaging was done at 37 °C and 5% $CO_2$. The cells were imaged every 15 or 20 min for 24 h. After imaging, the cells were removed by incubating in 0.05% trypsin for 1 h, and images of the fluorescent particles in the substrate were captured for a traction-free reference state for traction force microscopy[29]. To compute tractions, Fast Iterative Digital Image Correlation[30] was used using 32 × 32-pixel subsets (21 × 21 µm²) with a spacing of 8 pixels (5.2 µm) followed by Fourier transform traction microscopy[31] accounting for finite substrate thickness[6,32]. The displacements were computed Stresses within the monolayer were computed with monolayer stress microscopy[28,33,34]. Representative data showing images of the cells, fluorescent particles, tractions, and stresses are given in the SI, Fig. S5. Cell orientations were determined using the ImageJ plugin OrientationJ. Cell aspect ratios were quantified by the ratio of maximum and minimum eigenvalues of the structure tensor used by OrientationJ[35].

### Imaging and analysis for fixed cells

Imaging actin stress fibres required fixing the cell monolayers, which had to be done after collecting data for traction and stress measurements[36,37]. To this end, reference images of the fluorescent particles were collected before seeding the cells. Polyacrylamide gels were made as described above, and reference images of the fluorescent particles in the reference (undeformed) state were collected. Microscopy was the same as described above, with the exception of using a 20 × numerical aperture 0.5 objective (Nikon). MDCK cells were seeded and allowed to come to confluence overnight. The cell culture

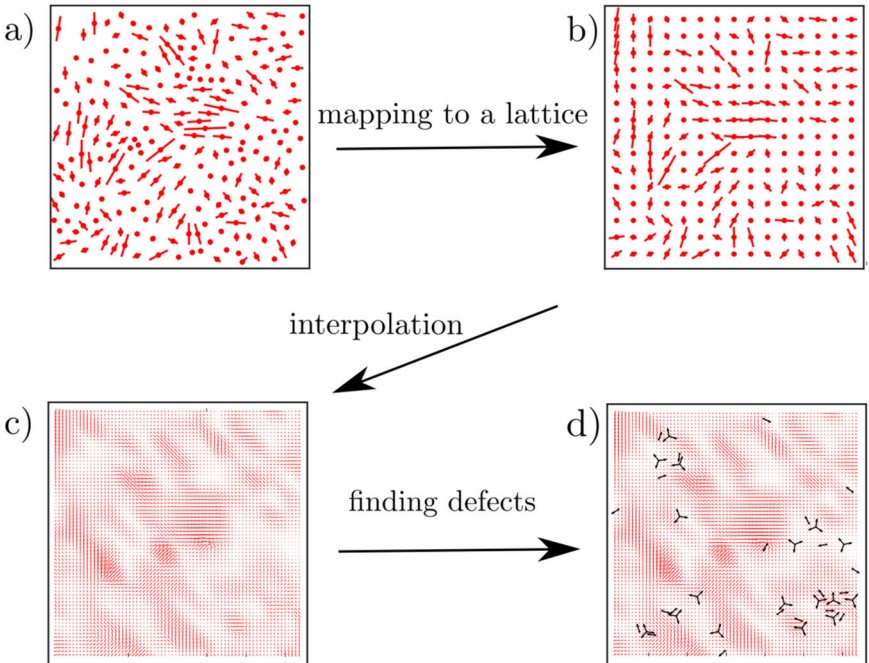

**Fig. 5 | Construction of the director fields for cell orientation and contractile stress. a** Cell position and orientation, as shown in Fig. 1. **b** Cell position and orientation after mapping to a lattice. The lattice unit is equal to $dl = L/\sqrt{N}$, where $L$ is system size and $N$ is the number of cells in the snapshot. **c** Linear interpolation is used to find the director field on a lattice with a smaller mesh. This step is required to find the position and orientation of defects correctly. **d** Using the director on the new lattice, we then use the defect-identifying algorithm introduced in ref. 39 to find the direction and orientation of ±1/2 defects.

medium was changed 1 h prior to imaging, and cells were imaged every 10 min for 1 h. Cell monolayers were fixed using chilled 4% paraformaldehyde solution for 20 min. Cells were then stained for actin using ActinRed 555 ReadyProbe Reagent (Invitrogen, catalogue number R37112) according to manufacturer instructions and images. To analyse the orientation of the fixed cells, the cells were manually segmented based on images of the cell cortex. The orientation of actin stress fibres was identified manually for cells having visually clear stress fibres. Stresses were computed using monolayer stress microscopy, as described above, but for these experiments, the stresses at the boundaries were unknown, meaning that the recovered stresses represented not the full stress tensor but rather deviations from the average of the stress tensor at the boundaries. Our prior experiments[28,38] and representative data (SI, Fig. S5) with this cell type show that the maximum shear stress is much smaller than the average principal stress, indicating the stress tensor is nearly isotropic. If the stress state at the boundary is isotropic, then there is no error in computing the orientation of principal stresses and the traceless stress tensor, both of which are reported in the main text for experiments using fixed imaging.

### Construction of the director fields for cell orientation and contractile stress

To analyse the experiments and simulations, we used Matlab 2022b. Figure 5 summarises the process of defining smooth director fields and identifying the position and orientation of defects[39]. First, cell positions and orientations are mapped onto a square lattice with a lattice constant $dl = L/\sqrt{N}$, where $L$ and $N$ are the system length and number of the cells in the snapshot, respectively. To find the orientation of the cells on a lattice, each cell is mapped to the closest lattice site. The result is shown in Fig. 5b. In the original lattice built by the cells, the lattice spacing is large and that makes it impossible to find the position and orientation of the defects accurately. As a result, we need to find the cell orientation on a lattice with a smaller mesh size. We construct this lattice by linear interpolation between

lattice points (Fig. 5c). Using the interpolated director field, we can then use a defect finding algorithm to find the position and orientation of ±1/2 defects (Fig. 5d).

The stress matrix measured in experiments has three independent components $\sigma_{xx}$, $\sigma_{xy}$ and $\sigma_{yy}$. It has a non-zero trace, and we first make it traceless by adding a constant $c$ to the diagonal elements, so that $\sigma_{yy} + \sigma_{xx} + 2c = 0$. To find the orientation of the contractile stress, we find the two mutually perpendicular axes which are parallel to the orientations of positive and negative principal stress. These axes can be found by a rotation of the stress matrix through an angle $\theta_p$ such that the shear stress $\sigma_{xy}$ becomes zero. We note that there are two directions over which the shear stress becomes zero, $\theta_p$ and $\theta_p + \pi/2$. We define the orientation of contractile cell-generated stress to be along the orientation of the positive stress (pointing outwards).

### The continuum model for active cell monolayers

We describe the motion of cells within the monolayer by a continuous velocity field **u**. The local orientation of cell shapes is described by a tensor field $\mathbf{Q}^n = S^n(\mathbf{nn} - \mathbf{I}/2)$ and the orientation of the stress fibres by a second tensor field $\mathbf{Q}^m = S^m(\mathbf{mm} - \mathbf{I}/2)$. These nematic order parameters encode the magnitude of nematic order in the cell shape, $S^n$, or the magnitude of the contractile stress, $S^m$, and the director field associated with the cell shape, **n**, or the stress, **m**[25]. This description differs from previous active nematic continuum theories in that it allows for a finite misalignment angle $\theta = \cos^{-1}(\mathbf{n} \cdot \mathbf{m})$ between the elongation axis of cell shape and the axis along which contractile forces are generated by stress fibres (see Fig. 2a).

Following empirical arguments, we assume that in equilibrium, the cell monolayer is governed by the following free energy density:

$$f = \frac{C}{2}(1 - 3\mathbf{Q}^n : \mathbf{Q}^n)^2 + \frac{C}{2}(1 - 3\mathbf{Q}^m : \mathbf{Q}^m)^2 + \frac{K_n}{2}|\nabla\mathbf{Q}^n|^2$$
$$+ \frac{K_m}{2}|\nabla\mathbf{Q}^m|^2 + \frac{J}{2}(1 - 3\mathbf{Q}^n : \mathbf{Q}^m)^2. \qquad (3)$$

In the absence of activity, the first and the second terms lead to a phase with nematic order in cell shape and cell stress. The third and the fourth terms penalise gradients in the shape and stress orientations, respectively. These are motivated by the observations of nematic ordering in both **n** and **m** in the experiments. The final term tends to align shape and stress orientations.

The shape orientation, **n**, and the stress orientation, **m**, change in response to active flows. Since they have different elastic constants, they respond differently to the active flows, which can lead to a mismatch between their directions. The dynamics of the nematic tensors is governed by[40]

$$(\partial_t + \boldsymbol{u} \cdot \boldsymbol{\nabla})\boldsymbol{Q}^n = -\boldsymbol{\Omega} \cdot \boldsymbol{Q}^n + \boldsymbol{Q}^n \cdot \boldsymbol{\Omega} + \gamma \boldsymbol{H}^n, \tag{4}$$

$$(\partial_t + \boldsymbol{u} \cdot \boldsymbol{\nabla})\boldsymbol{Q}^m = -\boldsymbol{\Omega} \cdot \boldsymbol{Q}^m + \boldsymbol{Q}^m \cdot \boldsymbol{\Omega} + \gamma \boldsymbol{H}^m, \tag{5}$$

where $\gamma$ is the rotational diffusivity, $\boldsymbol{\Omega} = (\boldsymbol{\nabla}\boldsymbol{u}^T - \boldsymbol{\nabla}\boldsymbol{u})/2$ is the fluid vorticity, the molecular field $\boldsymbol{H}^x = -(\frac{\delta f}{\delta \boldsymbol{Q}^x} - \frac{\boldsymbol{I}}{2} \operatorname{Tr} \frac{\delta f}{\delta \boldsymbol{Q}^x})$ shows the relaxation of the orientational order to the minimum of the free energy, and we have set the flow tumbling parameter equal to zero.

We assume that the flows observed in confluent cell monolayers can be well approximated by

$$\rho(\partial_t + \mathbf{u} \cdot \boldsymbol{\nabla})\mathbf{u} = \boldsymbol{\nabla} \cdot \boldsymbol{\Pi}, \tag{6}$$

where $\rho$ is mass density and the stress tensor $\boldsymbol{\Pi} = \boldsymbol{\Pi}_{\text{passive}} + \boldsymbol{\Pi}_{\text{active}}$ includes a passive and an active contribution, where the passive, viscous terms $\boldsymbol{\Pi}_{\text{passive}}$ are well known from liquid crystal hydrodynamics[40–42]. Flows in confluent cell layers are predominantly driven by active dipolar forces created on the single-cell level by stress fibres which convert chemical energy into mechanical work. This gives an active term in the stress tensor,

$$\Pi_{\text{act}} = -\zeta \mathbf{Q}^m. \tag{7}$$

where we choose $\zeta < 0$ to correspond to contractile forces.

We have assumed nematic ordering. Measurements of the cell aspect ratio in our experiments confirm that most cells have an elongated shape (see SI, Fig. S2a). It has recently been shown in experiments and numerical simulations that cell monolayers have nematic order at large length-scales and hexatic order at small length-scales[43,44] and it would be of interest in future work to explore any effect of any hexatic ordering.

The equations are solved using a hybrid lattice-Boltzmann algorithm[41,42]. The MDCK simulations are performed in a $200 \times 200$ box with periodic boundary conditions over 120,000 lattice-Boltzmann time-steps, and data is collected every 300 time-steps. The measurements are performed in a steady state when the mean number of defects and the fraction of the extensile area do not, change over time. The initial orientation of both **n** and **m** is random, and the magnitude of the order is $S^n = S^m = 1$. We choose values of parameters that lead to an active fluid in a low Reynolds number regime: $\rho = 40$, $\eta = 20/3$. Other parameter values are: $\gamma = 0.4$, $K_m = 0.005$, $C = 10^{-3}/3$, $\zeta = -0.03$, $K_n = 0.065$, $J = 0.0008$. We set up the LP-9 simulations such that nematic order only forms inside a circular region with radius $R = 80$. The free energy in this region is again given by Eq. (3), but the bulk free energy outside the circle is

$$f_{\text{bulk}} = \frac{C'}{2}(\boldsymbol{Q}^n : \boldsymbol{Q}^n)^2 + \frac{C'}{2}(\boldsymbol{Q}^m : \boldsymbol{Q}^m)^2 \tag{8}$$

which sets the magnitude of the order to be zero. We impose a defect in the shape director **n** at the centre of the inner region by setting the director angle with the x-axis to be equal to $\phi/2$, where $\phi$ is the polar angle in the co-ordinate system centred at the defect core. We do not allow **n** to vary in time; **m** relaxes towards the minimum of the free energy. We use the same parameter values as for the MDCK cells except $K_m = 0.02, K_n = 0.01, \zeta = 0, C' = 0.003$.

We note that the quantitative fits achieved using this model must be viewed with some caution as there are several adjustable parameters. However, the qualitative behaviour is insensitive to the numerical values of the parameters (SI, Fig. S1).

All the data used to plot the graphs are provided as Source data in ref. 45.

### Reporting summary

Further information on research design is available in the Nature Portfolio Reporting Summary linked to this article.

## Data availability

Source data are provided in this paper in ref. 45. The Source file includes the data presented in all the graphs. The name of each file inside the folder is the same as the figure it is representing. The name of the data in each file is the same as the name of the variable in the graph, and their standard deviation is specified by SD. Source data are provided with this paper.

## Code availability

The code used to analyze experimental data is available at https://github.com/jknotbohm. Simulation codes are available from the corresponding authors upon reasonable request.

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

## Acknowledgements

M. R. Nejad acknowledges the support of the Clarendon fund scholarship. L. J. Ruske acknowledges the support of the European Commission's Horizon 2020 research and innovation programme under the Marie Sklodowska-Curie grant agreement No 812780. This project was funded by the National Science Foundation grant number CMMI-2205141.

## Author contributions

M.R.N., L.J.R., G.Z. and J.M.Y. formulated the model and interpreted the results. M.R.N., M.M., J.Z. and J.N. analysed the experimental data. M.R.N. performed the simulations. J.N. designed and supervised the experiments. M.M. and J.Z. performed the experiments. LJ.R., M.R.N. and J.M.Y. drafted the manuscript. All authors commented on the manuscript.

## Competing interests

The authors declare no competing interests.
