## [Peer Review File · Nature Communications]

Stress-shape misalignment in confluent cell layersREVIEWER COMMENTS

Reviewer #1 (Remarks to the Author):

The authors report new experimental observations and numerical simulations of dynamical clusters of extensile cells surrounded by contractile in confluent cell tissues. Experiments on MDCK cells (more isotropic) and mesothelial cell line LP-9 (more elongated) are compared with numerical simulations of an adapted hydrodynamic model of nematics.

Extensile/contractile cells are identified by the relative angle between the principal axis of cell elongation (n director) and the principal axis of the contractile stress (m director). Most cells tend to generate contractile forces along their elongation axis, hence the relative angle between these two axes is close to zero. This is because the actomyosin bundles within each cell tend to be distributed along the long cell axis and are the primary source of contractile stress. Extensile cells are identified as those with large misalignments between the cell principal elongation axis and contractile forces, whereby cells tend rather to be compressed along their elongation axis.

These are interesting results that reveal a more complex, subtle interplay between the cell shapes and contractile stresses at the collective (tissue) level.

a) The authors make a crucial working hypothesis that active flows are the main source of stress-shape misalignment. This may be indeed important but is rather challenging to validate experimentally, and other sources such as cell motility, deformability and shape anisotropy which can be quantified experimentally are disregarded in the choice of modelling approach and analysis presented.

Active flows are constructed in theoretical models as a gradient flow of the active stress determined by nematic ordering. This is also done in the present study in a hydrodynamic model where two nematic order parameters are constructed one for the cell shape elongation (Q_n) and one for the principal axis of the contractile stresses (Q_m). The active flow is determined by the gradients in Q_m with the proportionality constant given by activity parameter as in conventional active nematics. This active flow is coupled to cell advection and changes in the nematic shape alignment. Numerical simulations of this model reveal that similar clustering of extensile cells are observed as in experiments. However, since active flows are not something measured in the experiments this is not a proof that active flows are the primary sources in the formation of extensile clusters.

Thus, even though this is an experimental study the working hypothesis on the active flow is not substantiated by experiments, which makes the interpretation of the results open to criticism.

b) The formation of clusters of extensile cells is attributed to active flows that somehow disturb the distribution of actomyosin bundle within the cells. Do the authors have experimental evidence of this working hypothesis? How would this be tested further? Is the idea that the cell activity (i.e. motility) can turn contractile cells into extensile ones? Or, is it that, at the collective level, active flow patterns can induce contraction of the cell shapes along their principal axis, thus they become "extensile"? This

working hypothesis needs to be further explained. No clear mechanism/process for the formation of extensile-like cells is put forward in a convincing way.

c) The profiles of the time correlations C_n and C_m are different in experiments and simulations (fig 3 b). The correlation of the shape elongation (C^n) obtained in simulations falls outside of the error bars from the experimentally obtained correlation function. This suggests, if anything, that the timescale for persistent shape alignment is rather different in simulations than in experiments. Thus, the statement on pg 5 lines 111-112 that:

“From the time correlation functions, in Fig. 3b, we identify a time-scale for the decay of the extensile patches ~ 300 minutes”

is ambiguous and puzzling. This should be explained and clarified.

d) The cell monolayers are fixed so the visualized active flows are “frozen” in time and due only to the stress fibers. However, it is conceivable and perhaps likely plausible that additional flows due to the actual cell motility would be competing to these active flows induced by cellular structures and would also play an important role in shaping cells. Have the authors also considered the effect of cell motility on the stress-shape misalignment?

e) The stress-shape misalignment is less pronounced in the experiments with the LP-6 cells which have a higher degree of shape elongation than the MDCK cells. Interestingly, the authors attribute this behavior to less pronounced active flows present in these tissues compared to the MDCK tissues. But this conclusion is not substantiated and seems rather biased on promoting the working hypothesis on active flows. It does not account for the effect of cell deformability and shape anisotropy which may be as important as the active flows. The authors need to provide more convincing evidence that indeed active flow is the main actor in misaligning shape orientation with respect to contractile stresses.

In view of all comments above, I believe that the paper needs to be substantially revised for further consideration for publication.

Reviewer #2 (Remarks to the Author):

Nejad et al describe in their MS a misalignment of cell shape with respect to traction towards the substrate in a confluent epithelial. As noted by the authors, for cells that typically live as individuals (like fibroblasts) the cell shape and the traction are highly correlated in their orientation (unfortunately the refs given in the MS, relate to late reviews rather than to some original papers, like those from Gardel, Schwarz and others).

Clearly, epithelial cells like MDCK within a confluent monolayer investigated in the current study behave different. In that situation cells need to balance the forces towards all neighbours, hence their shape is very different and traction forces will report on only a fraction of all the forces involved.

In this situation implying a nematic symmetry does not appear obvious. Even more strongly, Giomi's lab has shown in a series of papers earlier this year, that orientational order at the single-cell length scale in epithelia is rather hexatic, and not nematic. Just at length scales clearly beyond individual cells, at which also flow/mobility fields emerge, the organization turns to nematic order. Probably the traction field will align to those. I am surprised that the authors seem not to be aware of this novel description, or at least explain why such a model is not applicable to the experiments presented.

Certainly, there will be some correlation between (large scale) structural organization, and traction on the respective length scale in epithelial layers. The amount of which will be determined by a ratio of isotropic-to-nematic elastic coupling, analogous to that proposed in the MS. Likewise models have been developed for individual (fibroblast) cells earlier.

In general, the authors present potentially interesting data from which some new correlation between shape/organization and stress in epithelial monolayers could be extracted. Having the data, the authors should return to the drawing board and implement the novel developments by other groups, or clearly state why those do not apply.

Some further observations

- Fig.1e: it is unclear how the nematic directors were determined, and whether it complies with what one sees. There are spots where no director was assigned, yet a cell is clearly visible; there are dots (center-of-mass) right at the boundary of two cells. I wonder whether cell-segmentation is stable.

- Fig.3: error-bars in figures. It is not sufficient to consider error-bars between experiments, as shown, but also consider the considerable error made for each of the angles extracted.

- In eq(1) there seems a bracket missing

- p6: it is completely unclear where the cut-off length of 5.2 μ m comes from.

- TFM: needs to be quantified what's the density of particles, from there what is the predicted spatial resolution, show an example image and respective TFM result

- image analysis: it is unclear why cell data first are mapped onto a regular grid, and why even then needed to be (linearly?) interpolated, to obtain a pseudo 'high'-resolution image. Those kind of data-massage should be avoided. One should stick to the instrument's resolution and Nyquist's theorem.

- methodology: I am missing clear statements on the characterization of the methodology: what are the experimental accuracies in position and orientation, and how do they comply to the predictions?

We thank the reviewers for their helpful comments. In the reply the text in *italic blue* is the comments from the reviewers, that in black is our responses, and that in red highlights revisions to the manuscript.

REFeree 1

The authors report new experimental observations and numerical simulations of dynamical clusters of extensile cells surrounded by contractile in confluent cell tissues. Experiments on MDCK cells (more isotropic) and mesothelial cell line LP-9 (more elongated) are compared with numerical simulations of an adapted hydrodynamic model of nematics. Extensile/contractile cells are identified by the relative angle between the principal axis of cell elongation (n director) and the principal axis of the contractile stress (m director). Most cells tend to generate contractile forces along their elongation axis, hence the relative angle between these two axes is close to zero. This is because the actomyosin bundles within each cell tend to be distributed along the long cell axis and are the primary source of contractile stress. Extensile cells are identified as those with large misalignments between the cell principal elongation axis and contractile forces, whereby cells tend rather to be compressed along their elongation axis. These are interesting results that reveal a more complex, subtle interplay between the cell shapes and contractile stresses at the collective (tissue) level.

a)The authors make a crucial working hypothesis that active flows are the main source of stress-shape misalignment. This may be indeed important but is rather challenging to validate experimentally, and other sources such as cell motility, deformability and shape anisotropy which can be quantified experimentally are disregarded in the choice of modeling approach and analysis presented.

Active flows are constructed in theoretical models as a gradient flow of the active stress determined by nematic ordering. This is also done in the present study in a hydrodynamic model where two nematic order parameters are constructed one for the cell shape elongation (Q_n) and one for the principal axis of the contractile stresses (Q_m). The active flow is determined by the gradients in Q_m with the proportionality constant given by activity parameter as in conventional active nematics. This active flow is coupled to cell advection and changes in the nematic shape alignment. Numerical simulations of this model reveal that similar clustering of extensile cells are observed as in experiments. However, since active flows are not something measured in the experiments this is not a proof that active flows are the primary sources in the formation of extensile clusters.

Thus, even though this is an experimentally study the working hypothesis on the active flow is not substantiated by experiments, which makes the interpretation of the results open to criticism.

Cell motion in the monolayer (which is a consequence of cell activity) is equivalent to active flows in the continuum theory. We have now measured the cell velocity in the experiments. In Fig. 1 below we compare the vorticity field in snapshots of simulations and experiment, and more quantitatively, **the velocity-velocity correlation function has been added to the manuscript (Fig. 2 below)**. In particular, using the length-scale we found from matching the director correlation in experiments and in simulations, we can now compare the velocity correlation in experiments and simulations. We do not expect a perfect fit as we are using a continuum model that accounts for the underlying physics, rather than details of the biology. However, the decay in the velocity field in experiments show a similar behaviour.

There must also be a reason that the shape and stress dynamics respond differently to the active stress. In the model this is provided by the different elastic constants which cause a different response to the activity. The experimental evidence behind this assumption is the different length scales for the decays of the spatial correlation functions of m and n . The concept of an elastic constant is borrowed from equilibrium statistical physics, describing the propensity of elongated particles to align. In the context of cells it will depend on both deformability and shape anisotropy in ways that are not yet understood. So these are included in the model, but in an indirect way.

b)The formation of clusters of extensile cells is attributed to active flows that somehow disturb the

FIG. 1: An example of velocity patterns in experiments (a) and simulations (b). The background color shows the vorticity, black arrows show velocity and $+1/2$ and $-1/2$ defects are shown in magenta.

FIG. 2: (a) Decay of the spatial correlation functions, $C^x(r) = \langle \cos 2[\psi_x(r+r_0, t_0) - \psi_x(r_0, t_0)] \rangle_{t_0, r_0}$, where ψ_x represents shape director angle C^n , stress angle C^m , or the mismatch angle C^θ . Comparison of the simulation and the experiment assumes $10LB$ spatial units $\sim 30\mu m$. (b) Decay of the time correlation functions, $C^x(t) = \langle \cos 2[\psi_x(r_0, t+t_0) - \psi_x(r_0, t_0)] \rangle_{t_0, r_0}$, where ψ represents shape director angle C^n or the stress angle C^m . Comparison of the simulation and the experiment assumes $100LB$ time units ~ 10 min. (c) Using the length-scale that relates experiments and simulations (in part (a)), we can compare the velocity correlation in experiments (solid line) and simulations (dashed line). The velocity correlation and the error bars are measured in 8 experiments. (d) Magnitude of the stress σ as a function of the misalignment angle θ . The magnitude of the stress is re-scaled with its maximum value in each experiment before averaging. In (a), (b), and (d) the error bars show the standard deviation over 11 experiments.

distribution of actomyosin bundle within the cells. Do the authors have experiment evidence of this working hypothesis? How would this be tested further? Is the idea that the cell activity (i.e. motility) can turn contractile cells into extensile ones? Or, is this that, at the collective level, active flow patterns can induce contraction of the cell shapes along their principal axis, thus they become “extensile”? This working hypothesis needs to be further explained. No clear mechanism/process for the formation of extensile-like cells is put forward in a convincing way.

The reviewer is correct that the exact way in which the directors misalign in the experiments is not clear. The model assumes that the flows rotate the directors to change the cells from contractile to extensile. This could also happen in the experiments, but another possibility is that the cells shrink to circular and then re-extend in a perpendicular direction. A visual inspection of the data suggests that the transition is mostly happening by rotation rather than deformation.

To check this more quantitatively we have now measured cell aspect ratio as regions changed from contractile to extensile. The time origin is moved so that each region considered is contractile at $t=0$ and becomes extensile at $t=15$ min. The results show some change in cell shape during the transition, but it is not large. We cannot rule out larger changes in shape on a shorter time scale but do not observe these visually.

FIG. 3: a) Average cell aspect ratio right before and after formation of extensile patches. The changes in the aspect ratio are small. b) Average aspect ratio of the cells in 11 different experiments in extensile (blue) and contractile (red) regions ($p = 0.0003$, rank sum test). The data in (a) and (b) is averaged over 11 experiments.

We have also measured the cell aspect ratio in the contractile and extensile regions. There is a small reduction in the extensile regions which is consistent with Fig. 3d (which shows a smaller stress in the extensile regions).

We have discussed these points more fully in the paper.

c) The profiles of the time correlations C_n and C_m are different in experiments and simulations (fig 3 b). The correlation of the shape elongation (C^n) obtained in simulations falls outside of the error bars from the experimentally obtain correlation function. This suggests, if anything, that the timescale for persistent shape alignment is rather different in simulations than in experiments. Thus, the statement on pg 5 lines 111-112 that: “From the time correlation functions, in Fig. 3b, we identify a time-scale for the decay of the extensile patches ~ 300 minutes” is ambiguous and puzzling. This should be explained and clarified.

There is now considerable evidence that many of the phenomena seen in active nematics are similar to those observed in motile cell monolayers. However, up to now all active nematic models have not distinguished shape and stress directors. The novel point we are making is that if we allow shape and stress directors to respond to active flows in a different way we obtain the extensile patches seen in experiment and the localization of defects at interfaces. This is the case for a wide range of model parameters and assumptions. Fig 4 shows the appearance of patches and defects on boundaries for different parameter values. We have added this to the SM.

As is apparent from the replies to the reviewer’s first points, the exact mechanism for the director dynamics depends on biological detail and is complex. Therefore any quantitative match is due to a somewhat arbitrary tuning of parameters and should only be used to indicate that the mechanism is feasible. (Indeed we found that assuming a zero noise parameter in the model did not affect the quality of the fits and therefore, for simplicity, we have removed this parameter from the model description and this has resulted in very minor changes in the fits.)

$$K_n = 0.065 \quad \zeta = -0.03$$

$$J = 0.00008 \quad K_m = 0.005$$

$$J = 0$$

$$K_m = 0.01$$

$$K_m = 0.065$$

$$K_n = 0.08$$

$$K_n = 0.04$$

$$\zeta = -0.01$$

$$\zeta = -0.08$$

FIG. 4: The top left figure shows a snapshot of simulations for the parameter values used in the manuscript. The other sub-figures indicate that, for non-zero values of activity, extensile patches form for a wide range of values of the parameters in the simulations as long as cell shape and stress respond differently to the flows. In our model the difference comes from different elastic constants that lead to different spatial correlations ($C^n(r)$ and $C^m(r)$ in Fig. 2 of the reply). When cell shape and stress have the same elastic constant ($K_m = K_n = 0.065$), the cell shape and stress respond to flows in the same manner and extensile regions do not form.

We have reworded the comment on p5 to make it clearer: The physical values are from experiments and are used to define the length and time scale in the simulations.

d) The cell monolayers are fixed so the visualized active flows are “frozen” in time and due only to the stress fibers. However, it is conceivable and perhaps likely plausible that additional flows due to the actual cell motility would be competing to these active flows induced by cellular structures and would also play an important role in shaping cells. Have the authors also considered the effect of cell motility on the stress-shape misalignment?

To clarify, only the snapshots in Fig 1 are from fixed data. All the other results follow from live time-lapse experiments. Moreover “active flows” represent all (coarse-grained) cell motion and “active stresses” include all the forces acting on the cells.

e) The stress-shape misalignment is less pronounced in the experiments with the LP-6 cells which have a higher degree of shape elongation than the MDCK cells. Interestingly, the authors attribute this behavior to less pronounced active flows present in these tissues compared to the MDCK tissues. But this conclusion is not substantiated and seems rather biased on promoting the working hypothesis on active flows. It does not account for the effect of cell deformability and shape anisotropy which may be as important as the active flows. The authors need to provide more convicting evidence that indeed active flow is the main actor in misaligning shape orientation with respect to contractile stresses.

Our new measurements on the aspect ratio show that the aspect ratio of the cells does not dramatically change during the transition from contractile to extensile. This suggests that the transition is mainly led by cell rotation instead of cell deformation.

It is much harder for LP9 cells to turn the director of the longer cells or create new topological defects in the LP9 layer. Therefore the layer acts like a passive nematic and almost all defects anneal out (Duclos et al, Nat. Phys. 13, 58 (2017)). There are a few that remain and do not move. Near them the directors are stuck in extensile configurations.

The time correlation of the stress and cell shape presented below shows that the directors are almost stationary in time.

We have reworded the LP9 section to make this clearer.

In view of all comments above, I believe that the paper needs to be substantially revised for further consideration for publication.

We thank the referee for their helpful input which motivated new measurements which have improved the understanding of the dynamics.

REFeree 2

Nejad et al describe in their MS a misalignment of cell shape with respect to traction towards the substrate in a confluent epithelial. As noted by the authors, for cells that typically live as individuals (like fibroblasts) the cell shape and the traction are highly correlated in their orientation (unfortunately the refs given in the MS, relate to late reviews rather than to some original papers, like those from Gardel, Schwarz and others).

We are sorry to omit important references but unfortunately we are not sure what the reviewer is referring to here. The references about fibroblasts in paragraph 2 of the main text are to primary literature about nematic ordering in fibroblasts. We wondered whether the reviewer was concerned about the link between traction and shape in individual fibroblasts. As is apparent later in the paper

FIG. 5: Time correlations in the LP9 monolayer show that the cell shape and stress do not change appreciably in the experiment.

this is not very relevant (and may be misleading) because our study is about misalignment between cell shape and stresses within the plane of a cell layer as the reviewer notes below.

Clearly, epithelial cells like MDCK within a confluent monolayer investigated in the current study behave different. In that situation cells need to balance the forces towards all neighbours, hence their shape is very different and traction forces will report on only a fraction of all the forces involved.

In this situation implying a nematic symmetry does not appear obvious. Even more strongly, Giomi's lab has shown in a series of papers earlier this year, that orientational order at the single-cell length scale in epithelia is rather hexatic, and not nematic. Just at length scales clearly beyond individual cells, at which also flow/mobility fields emerge, the organization turns to nematic order. Probably the traction field will align to those. I am surprised that the authors seem not to be aware of this novel description, or at least explain why such a model is not applicable to the experiments presented.

Certainly, there will be some correlation between (large scale) structural organization, and traction on the respective length scale in epithelial layers. The amount of which will be determined by a ratio of isotropic-to-nematic elastic coupling, analogous to that proposed in the MS. Likewise models have been developed for individual (fibroblast) cells earlier.

In general, the authors present potentially interesting data from which some new correlation between shape/organization and stress in epithelial monolayers could be extracted. Having the data, the authors should return to the drawing board and implement the novel developments by other groups, or clearly state why those do not apply.

The work by Giomi's group is indeed interesting and novel. However very strong evidence that nematic ordering is relevant in our experiments is the existence of the non-zero nematic order parameter for both shape and stress and the existence of many topological defects with nematic symmetry. It is possible that there is also hexatic order on short length scales: at the cell densities used this is likely to be on the scale of one or a few cells and would not change our interpretation [Eckert, Nat Comms 2023].

As a check we have now measured the distribution of cell aspect ratios. It is not peaked around zero demonstrating elongated cells and the likelihood of nematic ordering.

We have contrasted the work by the Giomi group and added references to the manuscript.

Some further observations Fig.1e: it is unclear how the nematic directors were determined, and

FIG. 6: Distribution of aspect ratio in 11 different experiments over the entire time of the experiment. The peak is around 1.2 showing that the cells are not circular.

whether it complies with what one sees. There are spots where no director was assigned, yet a cell is clearly visible; there are dots (center-of-mass) right at the boundary of two cells. I wonder whether cell-segmentation is stable.

Fig. 1e was from an image of fixed cells and the orientations were done via manual cell tracing so the stability of the segmentation is not relevant in this case. We thank the referee for pointing out the errors which are due to a mistake in spatial re-scaling when plotting the data on top of the image.

We have corrected this.

- Fig.3: error-bars in figures. It is not sufficient to consider error-bars between experiments, as shown, but also consider the considerable error made for each of the angles extracted.

Any non-systematic errors in individual experiments will be included in the error bars we use. (If the systems were identical we would essentially be doing a standard error analysis of binning the data and seeing how much it compares from bin to bin.) Here we are also taking account of biological differences between systems which are expected to be considerably larger than experimental errors. The techniques we use are proven within the community thus lessening the likelihood of systematic errors.

- In eq(1) there seems a bracket missing

We checked and think it is OK.

- p6: it is completely unclear where the cut-off length of 5.2um comes from.

This is to some extent an arbitrary choice. It needs to be sufficiently large that we can reliably identify defects that are associated with a given interface but sufficiently small that the effect of the defect localisation at interfaces is not washed out. A sensible choice is $5.2 \mu m$, which is the spatial resolution of the measurements of stress. The prior manuscript did not specify this relationship to spatial resolution, and we have revised the manuscript to clarify this point.

- TFM: needs to be quantified what's the density of particles, from there what is the predicted spa-

tial resolution, show an example image and respective TFM result

The density of particles used was 0.036% weight/volume (where weight is the weight of the particles and volume is the volume of the polyacrylamide gel solution before it polymerized). This was reported in our prior manuscript that we cited in the methods section, and for clarity we have added it to the revised version. As for the question of spatial resolution, there is a tradeoff between spatial resolution and noise. For these experiments, we chose a spatial resolution and then verified that the noise was not excessive. To quantify noise, we computed the root mean square (rms) of tractions measured outside the cell layer. The ratio of rms traction produced by the cells to rms traction outside the cell layer was 12.0. **This information is included in a new supplemental figure (Fig 7 below), which also shows example images of cells, fluorescent particles, traction data, and stress data, as requested by the reviewer.**

- image analysis: it is unclear why cell data first are mapped onto a regular grid, and why even then needed to be (linearly?) interpolated, to obtain a pseudo 'high'-resolution image. Those kind of data-massage should be avoided. One should stick to the instrument's resolution and Nyquist's theorem.

To find the position and orientation of defects we use a method that is introduced in A. J. Vromans & L. Giomi, *Soft Matter* 12, 6490 (2016). To be able to systematically use this method to find the defects, we require the knowledge about orientation of the nematogens on grids. Mapping the data to a grid allows us to systematically calculate $\nabla \cdot Q$ which we require to find the orientation of the defects. The linear interpolation of the Q-tensor helps with identifying the position of defects as it allows one extra data point between the orientation measurement (theta) in the experiment. As a result, the magnitude of the nematic tensor Q can more easily reach low values around defects where the orientation changes a lot. There is nothing particular about linear interpolation – it only allows for the Q-tensor to more smoothly go to zero. Other types of interpolation will not crucially affect the position and orientation of defects. Note that the interpolation is not needed for the data analysis (correlation function, misalignment angle, ...) performed in this paper. It is only used for finding the position of defects.

Fig 8 shows how interpolation helps with identifying defects.

methodology: I am missing clear statements on the characterization of the methodology: what are the experimental accuracies in position and orientation, and how do they comply to the predictions?

We hope that the new information on methodology, in particular the added sample images from traction force microscopy and quantification of signal-to-noise ratio, has improved the manuscript. Additionally, as described above, we have fixed an error in scaling in Fig. 1 e-f, which we believe resolves the concern about accuracies in cell position and orientation. Thank you for the helpful comments.

FIG. 7: Representative data from traction force microscopy and monolayer stress microscopy. (a) Representative phase contrast image of an MDCK cell island. (b) Corresponding image of fluorescent particles in the substrate. (c, d) Components of traction (applied by the cells onto the substrate) in the x (c) and y (d) directions. (e) Magnitude of traction. (f, g) Stresses computed from monolayer stress microscopy, namely the average normal stress, $(\sigma_1 + \sigma_2)/2$ (f) and maximum shear stress $(\sigma_1 - \sigma_2)/2$ (g) with σ_1 and σ_2 being the principal stresses. (h) Orientation of first principal stress. Experimental noise in the traction data is quantified by computing the root mean square (rms) of traction in locations outside the cell layer, which, for this representative image, is 1.7 Pa. The rms traction produced by the cells is 20.6 Pa, giving a signal-to-noise ratio of 12.0.

FIG. 8: Cell orientation before (a) and after linear interpolation (b). The linear interpolation allows the Q-tensor to smoothly go to zero and as a result allows defect positions to be more easily identified.

REVIEWERS' COMMENTS

Reviewer #1 (Remarks to the Author):

The authors have addressed the comments and revised the manuscript in a satisfactory manner. Therefore, I recommend for publication.

Reviewer #2 (Remarks to the Author):

The authors replied/clarified their manuscript to my satisfaction.

There are some smaller glitches in Fig.2b, Fig.4c, Fig.7a.

In all cases they show probability-densities (not histogram!: 7a). As unit they give "%". In none of those figures integration leads to 100%; ie the pdf is wrongly normalized.

Still I'd like to see in eq(1): " $\cos(2x)$ " instead of " $\cos 2x$ "